# Efficacy and Safety of the Use of SGLT2 Inhibitors in Patients on Incremental Hemodialysis: Maximizing Residual Renal Function, Is There a Role for SGLT2 Inhibitors?

**DOI:** 10.3390/biomedicines11071908

**Published:** 2023-07-06

**Authors:** José C. De La Flor, Daniel Villa, Leónidas Cruzado, Jacqueline Apaza, Francisco Valga, Rocío Zamora, Alexander Marschall, Michael Cieza, Javier Deira, Miguel Rodeles

**Affiliations:** 1Department of Nephrology, Hospital Central Defense Gomez Ulla, 28047 Madrid, Spain; mroddel@oc.mde.es; 2Department of Nephrology, Clínica Universidad de Navarra, 31008 Pamplona, Spain; daniel.villa.hurtado@gmail.com; 3Department of Nephrology, Hospital General Elche, 03203 Elche, Spain; leocruzadov@gmail.com; 4Department of Nephrology, Hospital Fuensanta, 28942 Madrid, Spain; japaza77@gmail.com; 5Department of Nephrology, Hospital Universitario Doctor Negrin de Gran Canarias, 35016 Las Palmas de Gran Canarias, Spain; fvalga@hotmail.com; 6Department of Nephrology, Hospital Universitario General Villalba, 28400 Madrid, Spain; rociozgm@gmail.com; 7Department of Cardiology, Central Defense Gomez Ulla Hospital, 28047 Madrid, Spain; marschall.alexander@gmx.de; 8Teaching Coordination Unit, Universidad Peruana Cayetano Heredia, Lima 15012, Peru; michael.cieza@upch.pe; 9Department of Nephrology, Hospital San Pedro de Alcántara, 10003 Cáceres, Spain; deiralorenzo@gmail.com

**Keywords:** incremental hemodialysis, sodium/glucose cotransporter-2 inhibitor, residual kidney function, end stage renal disease, diabetic kidney disease, kidney replacement therapy

## Abstract

SGLT-2i are the new standard of care for diabetic kidney disease (DKD), but previous studies have not included patients on kidney replacement therapy (KRT). Due to their high risk of cardiovascular, renal complications, and mortality, these patients would benefit the most from this therapy. Residual kidney function (RKF) conveys a survival benefit and cardiovascular health among hemodialysis (HD) patients, especially those on incremental hemodialysis (iHD). We retrospectively describe the safety and efficacy of SGLT2i regarding RKF preservation in seven diabetic patients with different clinical backgrounds who underwent iHD (one or two sessions per week) during a 12-month follow-up. All patients preserved RKF, measured as residual kidney urea clearance (KrU) in 24 h after the introduction of SGLT2i. KrU levels improved significantly from 4.91 ± 1.14 mL/min to 7.28 ± 1.68 mL/min at 12 months (*p* = 0.028). Pre-hemodialysis blood pressure improved 9.95% in mean systolic blood pressure (SBP) (*p* = 0.015) and 10.95% in mean diastolic blood pressure (DBP) (*p* = 0.041); as a result, antihypertensive medication was modified. Improvements in blood uric acid, hemoglobin A1c, urine albumin/creatinine ratio (UACR), and 24 h proteinuria were also significant. Regarding side effects, two patients developed uncomplicated urinary tract infections that were resolved. No other complications were reported. The use of SGLT2i in our sample of DKD patients starting iHD on a 1–2 weekly regimen appears to be safe and effective in preserving RKF.

## 1. Introduction

Diabetic kidney disease (DKD) is the leading cause of end stage renal disease (ESRD) worldwide and continues to be the major contributor to kidney replacement therapy (KRT) [1,2]. Patients with DKD that develop macroalbuminuria have a greater risk of mortality due to cardiovascular disease (CVD) than they are to progress to ESRD [3]. Despite the significant decline in diabetes-related complications in recent decades, the same trend cannot be observed in chronic kidney disease (CKD) patients due to DKD that requires KRT [4]. Hence, there exists a significant requirement for novel treatment approaches that can enhance glycemic control while minimizing the risk of hypoglycemia, as well as reducing cardiovascular and renal risks within this population. Irrespective of the limitations associated with estimated glomerular filtration rate (eGFR), it is crucial to develop new treatments that can effectively address these concerns [5]. 

The American Diabetes Association (ADA) and Kidney Disease: Improving Global Outcomes (KDIGO) guidelines 2022 recommend sodium–glucose cotransporter-2 inhibitors (SGLT2i) as first-line treatment in all patients with an eGFR ≥ 20 mL/min/1.73 m^2^ and metformin in those with an eGFR ≥ 30 mL/min/1.73 m^2^ [6].

SGLT2i have become the new standard of care for slowing CKD progression in patients with type 2 diabetes mellitus (T2DM) [7,8,9] due to their specific renal and cardiovascular protective effects that are independent of the main metabolic and glucose-lowering effects [10,11]. Most studies on SGLT2i fail to include ESRD patients, particularly those on KRT or kidney transplant (KT) recipients [7,8,9]. These patients, due to their high risk of cardiovascular and renal disease, would benefit the most from these therapies. Literature on SGLT2i use in T2DM patients on KRT is limited. There is emerging evidence in post-hoc analyses and experimental and preclinical trials that support the premise that SGLT2i may be equally effective in preventing cardiovascular and mortality outcomes in patients on KRT, either on hemodialysis (HD), peritoneal dialysis (PD), or even in KT [12,13,14,15,16]. As of 2023, there are four major trials searching for a SGLT2i benefit in this population (NCT05687058, NCT05179668, NCT05141552, and NCT05374291).

In the majority of developed countries, the standard treatment for the vast majority of patients undergoing hemodialysis (HD) involves three sessions per week, each lasting 3–5 h. Regrettably, only a few centers adhere to the 2015 National Kidney Foundation-Kidney Disease Outcomes Quality Initiative (NKF-KDOQI) guidelines, which permit a reduction in the weekly HD dose for patients with a residual kidney urea clearance (KrU) exceeding 3 mL/min/1.73 m^2^ [17]. 

Incremental hemodialysis (iHD) has been proposed as an alternative to conventional HD, whereby dialysis dosage can be individually tailored according to RKF. In recent years, several randomized controlled trials (RCTs) and cohort studies have proved the safety and efficacy of iHD. Most of them developed once or twice-weekly HD regimens that showed no increased risk of mortality, fewer hospitalizations, and less cost when compared with standard HD treatment [18]. In these patients, preservation of residual kidney function (RKF) is associated with lower morbidity and mortality [17,19].

We report a case series on the efficacy and safety of SGLT2i treatment in patients with ESRD due to DKD on incremental hemodialysis with a regimen of 1–2 weekly sessions. In addition, we describe and evaluate the effects of SGLT2i on RKF during 12 months of follow-up.

## 2. Materials and Methods

Our study was carried out in the hemodialysis unit at Hospital Central de la Defensa Gómez Ulla in Madrid, Spain. We retrospectively collected information on all incident iHD patients from June 2021 until May 2023.

According to our current protocol, for once-a-week iHD, a patient must have a RKF measured as KrU of ≥4 mL/min/1.73 m^2^ and a urine volume of ≥1 L/day. This is based on the design proposed by Deira et al. in the IHDIP trial [20]. For a twice-weekly iHD, we follow the proposed criteria by Kalantar-Zadeh et al. [21]. Out of 23 patients starting iHD in the period described above, 12 patients (52.17%) had T2DM and 7 of these started treatments with SGLT2i (5 with Dapagliflozin and 2 with Empagliflozin) at a median of 3 months after starting iHD.

This is a retrospective analysis of those cases, and although treatment with SGLT2i has not been specifically approved for use in T2DM patients on HD, the off-label use is based on evidence from clinical trials that confirm that SGLT2i are cardioprotective and nephroprotective regardless of eGFR [7,8,9,10,11,12,22]. All patients signed their informed consent, and the reasons for off-label use of SGLT2i were clearly detailed in their medical records. Data on analytical parameters, RKF and HD, were collected at 0, 3, 6, 9, and 12 months.

Volemic parameters (bioimpedance spectroscopy, lung ultrasound (LUS), and other parameters for assessing systemic congestion) were collected only at 0, 6, and 12 months. Self-reported adverse events (gastrointestinal symptoms, genital fungal/urinary tract infections, and hypoglycemia events) and SGLT2i discontinuation were also recorded at these intervals.

## 3. Results

Table 1 presents the patient’s basal characteristics. The median age was 69.71 ± 10.24 years, and 85.7% were males. All were hypertensive, had renin–angiotensin system (RAS) blockade, and maintained furosemide treatment without changes during follow-up. ESRD etiology was DKD for all patients, but there was also secondary focal segmental glomerulosclerosis (FSGS) due to decreased renal mass after radical nephrectomy for renal cancer (Patient 1), right nephroureterectomy for high-grade (Patient 4), crescentic IgA nephropathy (IgAN) (Patient 5), and advanced IgAN (Patient 7).

Five patients started on a once-weekly iHD regimen, with two on a twice-weekly regimen. All patients received post-dilution online hemodiafiltration (OL-HDF) with a dialyzer of asymmetric cellulose triacetate (ATA) membrane and venous administration of low-molecular-weight heparin (LMWH, Enoxaparin) at a dose of 40 or 60 mg in each session.

A significant improvement in KrU levels was found. It increased from 4.91 ± 1.14 mL/min to 7.28 ± 1.68 mL/min (*p* = 0.028) at 12 months (Figure 1a). Urine volume increased from 1742 ± 288 mL/24 h to 2021 ± 532 mL/24 h, and 24 h creatinine clearance also increased from 12.7 ± 3.53 mL/min to 16.35 mL/min ± 6.85 (*p* = 0.26) but was not significant. Urine albumin/creatinine ratio (UACR) and 24 h proteinuria were also significantly reduced at the end of the study. UACR dropped from 4040 ± 2729 mg/g to 1568 ± 746 mg/g (*p* = 0.016) and 24 h proteinuria from 4.10 ± 1.95 g/24 h to 1.88 ± 0.71 g/24 h (*p* = 0.028) (Figure 1b).

The extracellular water to total body water ratio (ECW/TBW), by bioimpedance measurement, was significantly reduced at 12 months (Figure 2a). Weight, body mass index (BMI), and fat mass reduction were non-significant, as was the increase in lean mass (Figure 2b,c).

As shown in Table 2, a significant drop was observed in pre-hemodialysis blood pressure (both systolic and diastolic) at the end of the study. As a result, antihypertensive medication was modified. Calcium channel blocker (CCB) was discontinued in Patients 6 and 7, the same as angiotensin-converting enzyme inhibitors (ACEi) in Patients 3 and 5. Beta-blocker (BB) dose was reduced in Patients 5 and 7.

There was also a significant improvement in blood uric acid (from 7.67 ± 1.95 mg/dL to 5.96 ± 0.52 mg/dL, *p* = 0.018), serum potassium levels (from 5.45 ± 0.43 mmol/L to 4.62 ± 0.39 mmol/L, *p* = 0.001), serum phosphorus levels (from 5.7 ± 0.39 mg/dL to 4.68 ± 0.39, *p* = 0.001), 25-Hydroxyvitamin D (25OHD) serum levels (from 18.94 ± 4.85 mg/dL to 27.08 ± 2.02 mg/dL, *p* = 0.007), and hemoglobin A1c (HbA1c) levels (6.79 ± 0.68 g/dL to 6.36 ± 0.59 g/dL, *p* = 0.018) (Figure 2d). We observed no significant difference in calcium levels or parathyroid hormone (PTH) (*p* = 0.42 and 0.122, respectively). Of note, at the third month of starting treatment with dapagliflozin, Patients 4 and 7, who were treated with semaglutide and insulin, were able to discontinue (patient 4) and reduce their usual insulin doses by 20% (Patient 7).

Regarding side effects, two patients developed urinary tract infections (UTIs). Patient 3 developed a UTI after 3 weeks of treatment initiation with dapagliflozin. Patient 6 developed it after 8 weeks with empagliflozin. Both episodes were treated and resolved. SGLT2i discontinued in none of the patients.

No vascular access complications, change in mean KT or infusion volume in post-dilutional OL-HDF, or problems with blood flow rate (QB), arterial pressure flow (APF), or venous pressure flow (VPF) developed during the 12-month follow-up (Table 2).

## 4. Discussion

In this report, we describe the safety and efficacy of SGLT2i regarding RKF preservation in seven diabetic patients with different clinical backgrounds who underwent iHD (one or two sessions per week) during a 12-month follow-up.

SGLT2i, originally developed as oral hypoglycemic drugs, have shown renal-specific and cardioprotective effects to prevent the progression of CKD in numerous landmark cardiovascular outcome (CVO) trials [7,8,9]. Evidence is not clear whether these benefits can be extrapolated to patients with more advanced CKD or on dialysis, particularly those maintaining RKF on PD or iHD.

A post hoc analysis of The CREDENCE trial [7] described the efficacy and safety of canagliflozin in participants with eGFR below 30 mL/min/1.73 m^2^. In total, 174 individuals (4%) out of the total 4401 participants randomized for the trial had an eGFR below 30 mL/min/1.73 m^2^ at the time of randomization, with an average eGFR of 26 mL/min/1.73 m^2^. This analysis revealed that canagliflozin effectively slowed down the progression of kidney disease in advanced DKD patients without increasing the risk of acute kidney injury (AKI) [22]. Additionally, Chertow et al. [12] showed the beneficial effects of dapagliflozin on the reduction of renal and cardiovascular events, as well as the delay in the progression of eGFR decline in patients with an eGFR below 30 mL/min/1.73 m^2^, independently of the presence of type 2 diabetes (T2DM).

The recently completed EMPA-KIDNEY trial included patients with CKD (T2DM and without T2DM) who had an eGFR between 20 mL/min/1.73 m^2^ and 90 mL/min/1.73 m^2^ with UACR ≥ 200 mg/g. Patients were randomly assigned to receive empagliflozin (10 mg once daily) or a matching placebo. The primary outcome was a composite of the progression of kidney disease or death from cardiovascular causes. Of the 6609 patients that underwent randomization, 1131 (34%) in the control group and 1151 (35%) in the placebo group had an eGFR < 30 mL/min/1.73 m^2^. This trial demonstrated consistent benefits of empagliflozin treatment in reducing the risk of kidney disease progression or death from cardiovascular causes compared to the placebo, even in the population with lower eGFR [9].

Ongoing randomized clinical trials (RCT) should provide further evidence to support the usefulness of SGLT2i in HD. We look forward with optimism to the results of the following studies currently underway: (1) SGLT2i in hemodialysis (DAPA-HD). Examine the effect of dapagliflozin for left ventricular mass indexed to body surface area (LVMi) reduction as measured by cardiac magnetic resonance imaging at baseline and after 6 months of treatment in comparison with a placebo in patients undergoing replacement therapy with hemodialysis (NCT05179668). (2) Safety, tolerability, and feasibility of empagliflozin therapy in dialysis-dependent ESKD (EM-PA-HD). The aim is to include 75 diabetic and non-diabetic patients on dialysis for >3 months. They are randomized into three treatment arms: empagliflozin 10 mg, empagliflozin 25 mg, or placebo. The primary variable is the proportion of patients still on treatment at each dose at the end of 12 weeks of treatment. In the secondary variables, the trial will look at the adherence, safety, and pharmacokinetics of empagliflozin in these dialyzed patients (NCT05687058). (3) The RENAL LIFECYCLE trial is a pragmatic, randomized, controlled clinical trial with a basket design. It plans to enrol 1500 participants (including 450 to 525 patients with CKD stages G4/5, 400 to 475 patients on dialysis, and 550 to 650 patients with a KT), consisting of a screening period and a double-blind treatment period with two arms. The primary outcome measure is the combined endpoint of all-cause mortality, kidney failure, and hospitalization for heart failure in the overall study population. It should be noted that when pre-dialysis patients start dialysis or dialysis patients are transplanted, they will continue in the trial, and this will not be a reason to discontinue the study. Of note, the inclusion criteria for the dialysis patients specify that they must maintain a residual diuresis >500 mL/24 h at least 3 months after the start of dialysis (NCT05374291). In addition, the included dialysis patients should have a residual diuresis >500 mL/24 h at least 3 months after starting dialysis (NCT05374291). (4) The safety of dapagliflozin in hemodialysis patients with heart failure (SDHF) is an open, randomized, controlled study. The aim is to recruit 20 hemodialysis patients with heart failure. Among these participants, 10 individuals will receive a daily dose of dapagliflozin at 10mg for a duration of 12 weeks. The primary outcome measure focuses on determining the number of patients who experience hypoglycemia or urinary infection. The secondary outcome involves assessing the changes in NT-proBNP levels (NCT05141552). These and still developing trials support the premise that, even in the context of minimal diuresis and low SGLT2 receptor disposition, as expected in incident iHD patients who still maintain significant RKF, SGLT2i may exert favorable direct and indirect effects in preventing cardiovascular and mortality outcomes and other benefits such as the preservation of the RKF [13].

RKF conveys a survival benefit and cardiovascular health among HD patients. Clearance of protein-bound and middle molecules, reduction of inflammation, and improved fluid management are among the proposed mechanisms [23]. Therefore, preservation of RKF is associated with better patient outcomes, including survival and better quality of life [24,25]. There is no uniform definition for RKF, it can be estimated and measured, but an optimal method for it has not been established. The NKF-KDOQI guidelines advocate measuring RKF by calculating the residual kidney urea clearance (KrU) in 24 h urine and expressing it in mL/min/1.73 m^2^ in HD patients.

Preservation of RKF in iHD patients requires not only adequate BP, proteinuria, diuretics, and volume control but also avoiding intradialytic hypotension and continuous adjustment of hemodialysis prescription by measurement and monitorization of RKF [23].

In our center, the intensity of the iHD regimen is tailored based on residual kidney function (RKF). Patients may begin with a once-weekly regimen if their KrU is in the range of 4 to 5 mL/min/1.73 m^2^. If the KrU decreases to a range of 2 to 4 mL/min/1.73 m^2^, the regimen progresses to twice weekly. Finally, if the KrU falls below 2 mL/min/1.73 m^2^, the regimen is adjusted thrice weekly. This approach allows for individualized treatment based on the specific level of residual kidney function. 

The seven patients preserved creatinine clearance (CrCl) and KrU 24 h after the introduction of SGLT2i, and two of them (Patient 1 and 5) even improved their eGFR, making it possible to discontinue hemodialysis. In all cases except for Patient 7, KrU and CrCl increased with an average of 2.7 mL/min and 3.65 mL/min, respectively. Although CrCl decreased in this patient, urinary creatinine and urea excretion normalized to kilogram weight were maintained. A possible cause of the decrease in CrCl is the significant increase in lean mass and serum Cr.

Currently, there are no trials analyzing the role of SGLT2i in the preservation of RKF in patients with advanced DKD undergoing iHD. The KrU findings from our case series may suggest some additional beneficial effects of SGLT2i on RKF. Nevertheless, the importance of regularly monitoring RKF in all KRT modalities and adjusting dialysis prescriptions cannot be stressed enough.

KDOQI guidelines [17] recommend ACEi or ARB-II to control BP in patients with significant RKF either on PD or HD because this treatment was associated with an improvement in residual glomerular filtration rate (GFR) and urine volume on HD [26]. We know now that renin–angiotensin system (RAS) inhibitors continuation in non-dialysis advanced CKD patients is not associated with a significant loss of RKF in the long term [27]. Improvements in BP control are associated with hemodynamic stability during dialysis and with the maintenance of RKF and fewer cardiovascular events [28].

It is well known that the risk of hyperkalemia in patients with HD is high and multifactorial. It could be associated with long interdialytic intervals, acidosis, food intake, or medications such as ACEi/ARB-II or mineralocorticoid receptor antagonists (MRA). In our series, we observed a significant drop in the levels of potassium at the end of the follow-up. Although the exact mechanism is not clear, there is a hypothesis from a recent meta-analysis that SGLT2i may increase distal sodium and water delivery, enhancing the electronegative charge in the tubular lumen that regulates potassium excretion in the distal nephron caused on some occasions by the combined regimen including ACEi or MRA [29].

To reconcile available evidence, controversial results exist that the use of SGLT2 may have some deleterious effects on bone health in specific subgroups of patients at high risk for bone fracture [30]. This is associated with the combination of decreased 1,25-dihydroxyvitamin D (1,25(OH)2D) and increased PTH and fibroblast growth factor-23 (FGF-23) serum levels, which combined possibly contribute to the increased fracture risk associated with SGLT2i, but such results were not present in our series.

In our study, we observed a significant reduction of SBP after 12 months of SGLT2i treatment. Similar data were obtained in a cohort of 50 patients with T2DM and advanced CKD on automated peritoneal dialysis (APD) who had SGLT2i (dapagliflozin) added to their insulin therapy. Compared to SBP at the start of treatment versus 6 months after, the authors observed a significant decrease from 148 ± 5.2 mmHg to 134 ± 6.5 mmHg (*p* = 0.0431) [31]. Although modest, to our knowledge, these effects on BP may contribute to the potential reno-protective and RKF-preserving actions of SGLT2i. Studies in animal models demonstrate that SGLT2 inhibition causes sympathetic inhibition of renal nerve function, which could explain the glucose-independent effect of SGLT2 inhibition on BP [32]. The denervation hypothesis is consistent with the glucose-independent renal and cardiovascular benefits observed in the subgroup of patients with eGFR below 30 mL/min/1.73 m^2^ treated with canagliflozin [22], as well as in the DAPA-HD study with dapagliflozin [33]. Of note, there are contradictory data on the benefits of ACEi and ARB-II in reducing mortality and preserving RFK in patients on hemodialysis, independent of attained blood pressure. Xydakis et al. [26] conducted a study in 42 hypertensive patients on HD treated with enalapril, showing the association of increased preservation of RRF at 1-year follow-up. In contrast, Kjaergaard et al. [34] did not find the same benefit with irbesartan in HD patients. Patients in our series were already on these drugs before starting hemodialysis. Therefore, we hypothesized that SGLT2i has additional benefits when used concomitantly with RAS blockade, not only for the reduction in BP but also for its antiproteinuric effect despite extended glomerular damage in ESRD [35]. T2DM is a leading cause of proteinuria in ESRD patients, which is related to cardiovascular events [36]. However, persistent proteinuria is associated with a decline in RKF, independent of ESRD etiology [37]. RAS blockade and SGLT2i reduce albuminuria by decreasing intraglomerular pressure (SGLT2i by afferent arteriolar vasoconstriction, RAAS blockers by efferent arteriolar vasodilation). It is well known that one of the most relevant factors for the preservation of RKF is the reduction in urinary albumin excretion. Therefore, SGLT2i should have beneficial effects on RKF in patients with iHD and PD. In our case series, we observed significant albuminuria reduction, consistent with the findings of clinical trials performed in non-hemodialysis T2DM CKD patients [7,8,9].

During the 2022 KDIGO controversies conference on blood pressure and volume management in dialysis, reference was made to the fact that the balance between correcting chronic hypervolemia and preventing acute intravascular volume depletion remains a critical challenge in the care of HD patients. Various strategies, such as diuretics and volume control, have been proposed for volume management in HD patients, and this will depend primarily on the availability of accurate and objective methods to assess volume status. Clinical examination is currently the mainstay of volume assessment; however, this approach is inaccurate and unreliable. Other tools such as biomarkers, LUS, and bioimpedance designed to objectively aid in the assessment of ECW are currently being tested for efficacy and safety. In our hemodialysis unit, we use bioimpedance, an 8-zone LUS protocol and assessment of the inferior vena cava (IVC), portal, and hepatic vein to avoid high ultrafiltration rates and the risk of intradialytic cardiac stress, intradialytic hypotension (IDH), organ damage, and loss of RKF. Our results show that these tools are very useful in preserving RKF; however, more studies are needed to validate the efficacy and safety of these interventions in hemodialysis patients.

In addition, the positive effects of diuretic use in preventing excessive volume overload and preservation of RKF in HD patients are known. The Dialysis Outcomes and Practice Pattern Study (DOPPS) [38] showed the benefits of loop diuretics on better control of hyperkalemia, lower interdialytic weight gain, greater likelihood of preservation of RKF, and lower cardiac mortality below 14%. At the moment, it is unclear whether the diuretic effect of SGLT2i is attributable to osmotic diuresis or natriuresis, or both. SGLT2i can reduce intravascular volume status and thus preserve RKF. The records of our patients showed an increase in diuresis after starting the medication; these data agree with the results obtained in the study performed by Alhwiesh et al. [31] in T2DM patients with RKF in APD treated with SGLT2i.

Another relevant observation is that our patients did not require hospital admissions for heart failure; only one patient (Patient 4) required three extra hemodialysis sessions for volume adjustment. This low incidence of volemic complications could also be attributed to the strict control that we performed during the follow-up based on the point of care ultrasound (POCUS) and bioimpedance, as well as the constant periodical assessment by the nursing staff. The weight and ECW/TBW ratio of our patients decreased by a mean of 2.8 kg (*p* = 0.128) and 0.009 (*p* = 0.028), respectively. All these findings are similar to those found in patients with non-hemodialysis CKD.

High ultrafiltration rates (UFR) in HD patients are a risk factor for RKF loss [39]. There are a large number of observational studies demonstrating the association of higher UFR with all-cause and cardiovascular mortality. The findings of an observational study indicate that patients who experience a higher proportion of HD sessions with UFR exceeding 13 mL/h/kg during the initial three months of HD have an elevated risk of mortality. This increased risk persists even when the average UFR during that period remains below 13 mL/h/kg [40]. In our case series, interdialytic weight gain decreased by a mean of 0.39 L, with a consequent low UFR. We believe that SGLT2i use in incident iHD patients could help reduce interdialytic weight gain, avoiding high UFR rates, and preserving RKF.

Another important point is the effects of SGLT2i on body composition in patients with T2DM. Bioimpedance spectroscopy also measures other body composition parameters such as BMI, fat mass, and lean mass. In our case series, patients had a mean reduction in fat mass of 3 kg, with a mean increase in lean mass of 2 kg after starting SGLT2i, measured by bioimpedance. These findings are inconsistent with recent evidence pointing to an increased risk of sarcopenia by loss of skeletal muscle mass in diabetic patients treated with SGLT2i [41,42,43,44,45,46,47,48]. A recent meta-analysis in type 2 diabetic patients treated with SGLT2i showed a significant reduction in skeletal muscle mass compared to other antihyperglycemic agents [49]. The hypothesis put forward in this regard is that SGLT2i induces skeletal muscle loss to increase the release of amino acids into the systemic circulation as a catabolic response to renal glucose loss, preventing hypoglycemia [50,51]. Quiroga et al. [52], in a recent editorial, mention this issue, the effect of SGLT2i on skeletal muscle mass makes renal endpoints based on serum creatinine questionable (due to overestimation of eGFR), especially in cardiorenal patients, who per se present with muscle mass loss and sarcopenia. These aspects could influence the interpretation of the true renal and CV beneficial effects of SGLT2i. The authors recommend considering the use of muscle mass-independent estimates of eGFR, such as the calculation of eGFR based on cystatin C measurement. 

The inhibitor effects of the coupled reabsorption of sodium and glucose in the proximal tubule of the kidney from SGLT2i lead to natriuresis and glycosuria. Therefore, SGLT2i promote the renal excretion of glucose and modestly lower elevated blood glucose levels. In patients with T2DM and moderate renal impairment, it was assumed that the HbA1c-lowering efficacy and micro-macrovascular preventive action of SGLT2i are attenuated or absent due to a reduction in the number of functional nephrons in proximal tubules containing SGLT2 receptors [53]. Nevertheless, despite this reduction in the capacity for tubular glucose reabsorption, SGLT-2i have been shown to be safe in diabetic patients with mild to moderate CKD [54]. The reduced glucosuric effect associated with low eGFR was observed in our cases series, but despite this, we observed a significant reduction in HbA1c and increased glycosuria. We assume that this improvement is due to two of the seven patients receiving GLP-1-RA treatment and not because of the direct effect of SGLT2i since the magnitude of serum glucose reduction is known to be dependent on glycosuria. We observed a beneficial effect of SGLT2i regarding serum uric acid with a mean reduction of 1.71 mg/dL, which is in agreement with the study by Alhwiesh et al. [31]. A meta-analysis revealed a significant reduction in uric acid levels after treatment with any of the SGLT2i [55]. The possible mechanisms underlying the hypouricemic effect of SGLT2i have not yet been established.

Regarding side effects, SGLT2i treatment in our case series was safe. Two patients developed a single episode of UTI. The medication was not discontinued in any of them. No cases of euglycemic ketoacidosis, bone fractures, or amputations were reported, but it must be addressed that our sample was small, and the follow-up period may have been too short.

The main limitation of our study is the small sample size. The absence of a control group and the fact that all data were retrospectively collected should also be of note. However, our study, as far as we are concerned, is the first report describing the possible beneficial effects of SGLT2i on RKF preservation in T2DM patients on iHD.

## 5. Conclusions

The use of SGLT2i in our small sample of patients with ESRD and DKD starting iHD on a 1–2 weekly regimen appears to be safe and effective in lowering HbA1c, improving BP control, reducing proteinuria, serum uric acid levels, and interdialytic weight gain, and most importantly in preserving RKF.

Clinical data supporting the cardio and nephroprotective effects of SGLT2i are currently limited to patients with eGFR >20 mL/min/1.73 m^2^, but emerging evidence is expanding its use and indication. The necessity of more studies on the effects of SGLT2i on patients with advanced CKD, kidney transplants, and dialysis patients cannot be stressed enough. The findings in this case series need to be confirmed in a large prospective study with a longer follow-up period.

## Figures and Tables

**Figure 1 biomedicines-11-01908-f001:**
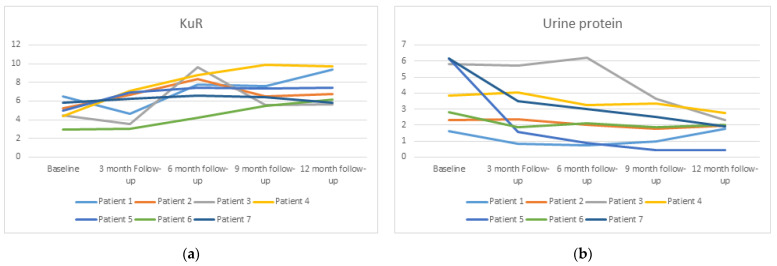
Change in residual kidney urea clearance (KuR) (**a**) and urine protein (**b**) over time during the data collection period of 1 year.

**Figure 2 biomedicines-11-01908-f002:**
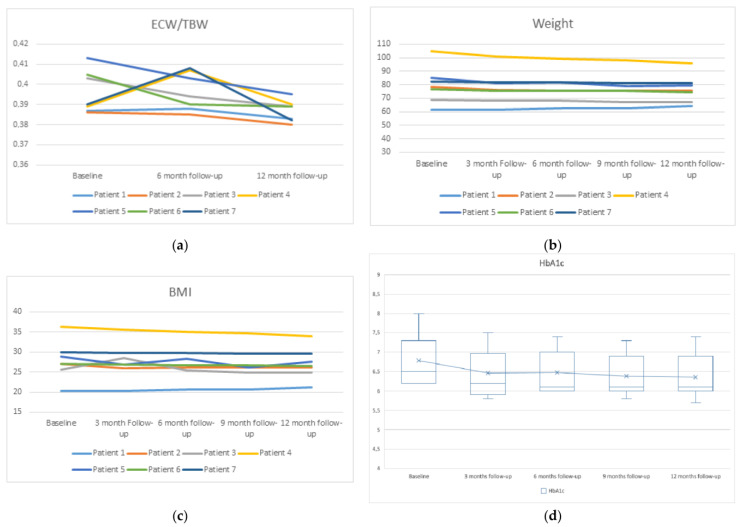
Changes in ECW/TBW (**a**), weight (**b**), BMI (**c**), and glycated hemoglobin A1c (HbA1c) (**d**) with respect to their baseline.

**Table 1 biomedicines-11-01908-t001:** Basal characteristics before starting SGLT2i.

	Patient 1	Patient 2	Patient 3	Patient 4	Patient 5	Patient 6	Patient 7
Age	82	64	62	80	64	79	57
Sex	Male	Male	Female	Male	Male	Male	Male
BMI	20.3	26.8	28.7	36.3	27.7	27.3	31.75
Hypertension	Yes	Yes	Yes	Yes	Yes	Yes	Yes
RAS blockade	ACEi	ARB	ACEi	ACEi	ACEi	ACEi	ARB
Mean Basal BP (mmHg)	150/70	166/85	140/78	135/84	128/79	168/78	156/94
Furosemide (mg/day)	120 mg/day	80 mg/day	120 mg/day	60 mg/day	120 mg/day	80 mg/day	120 mg/day
Cardiovascular Disease	No	No	Yes	Yes NVAF on DOACs	Yes NVAF on DOACs	No	Yes NVAF on LMWH
Type 2 Diabetes mellitus	Yes	Yes	Yes	Yes	Yes	Yes	Yes
History T2DM (years)	15	15	20	20	20	10	15
HbA1C	6.3%	6.2%	6.2%	6.5%	7.3%	7%	8%
Treatment	DPP4i	DPP4i Repaglinide	DPP4i Repaglinide	GLP-1-ra Degludec 8 IU	DPP4i	DPP4i	GLP-1-ra Determir 32 IU
SGLT-2i started	Dapagliflozin 10 mg	Empagliflozin 10 mg	Dapagliflozin 10 mg	Dapagliflozin 10 mg	Dapagliflozin 10 mg	Empagliflozin 10 mg	Dapagliflozin 10 mg
ESKD etiology: Diabetic kidney disease	Yes	Yes	Yes	Yes	Yes	Yes	Yes
Other	Secondary FSGS after nephrectomy	None	None	Right nephron-ureterectomy for high-grade urothelial carcinoma	Crescentic IgA Nephropathy	None	Advanced IgA Nephropathy
Dialysis vintage before starting SGLT2i	3 months	3 months	3 months	3 months	3 months	3 months	3 months
Vascular access type	aAVF	aAVF	aAVF	aAVF	Tunneled catheter	aAVF	aAVF
KuR	4.63	5.3	4.01	4.37	4.55	2.98	6.1
24 h CrCl (mL/min)	15	14.7	16	10	10	7.6	15.3
Residual diuresis (mL/24 h)	1400	1900	1900	1750	1900	1300	1900
Albumin/Cr (mg/g)	638	1696	5300	3750	3200	3440	4000
Proteinuria (g/24 h)	1.2	2.3	5.8	4.15	10	3.5	4.0

ACEi: angiotensin-converting enzyme inhibitors; ARB: angiotensin receptor blockers; BMI: body mass index; DDP4i: dipeptidyl peptidase IV inhibitors; SGLT-2i: sodium/glucose cotransporter-2 inhibitors; GLP-1-ra: glucagon-like peptide-1 receptor agonist; NVAF: non-valvular atrial fibrillation; DOAC: direct oral anticoagulants; LMWH: low molecular weight heparin; FSGS: focal segmental glomerulosclerosis. aAVF: autologous arteriovenous fistula; KrU: residual kidney urea clearance; 24 h CrCl: 24 h creatinine clearance.

**Table 2 biomedicines-11-01908-t002:** Evolution of dialysis parameters, residual renal function, bioimpedance, volemic parameters, and laboratory characteristics.

Variable	Baseline	3-Month Follow-Up	6-Month Follow-Up	9-Month Follow-Up	12-Month Follow-Up	*p*-Value *	*p*-Value **
Weight (kg)-mean (SD)	79.7 (13.7)	78.1 (12.4)	77.8 (11.8)	77.0 (11.4)	76.9 (10.4)	0.034	0.128
BMI (kg/m^2^)-mean (SD)	28.3 (4.99)	27.7 (4.62)	27.4 (4.40)	26.9 (4.34)	27.1 (3.96)	0.051	0.075
ECW/TBW	0.396 (0.01)	-	0.396 (0.009)	-	0.387 (0.005)	0.021	0.028
Fat mass (Kg)-mean (SD)	27.9 (12.8)	-	25.5 (11.3)	-	24.9 (10.6)	-	0.085
Lean mass (kg)-mean (SD)	47.6 (2.8)	-	48.9 (3.29)	-	49.6 (2.7)	-	0.049
LUS (N° B lines)-mean (SD)	8.43 (3.45)	-	7.71 (2.42)	-	5.28 (1.70)	-	0.031
PV PF (%)-mean (SD)	32.5 (7.5)	-	27.6 (4.4)	-	26.4 (4.7)	-	0.044
Hepatic vein							
S > D at HVF-N (%)	5 (71.4)	-	6 (85.7)	-	7 (100)	-	-
S < D at HVF-N (%)	2 (28.5)	-	1 (14.2)	-	0 (0)	-	-
S Reversal at HVF-N (%)	0 (0)	-	0 (0)	-	0 (0)	-	-
IDWG (kg)-mean (SD)	0.97 (0.44)	-	0.7 (0.26)	-	0.58 (0.21)	-	0.009
KT (L)-mean (SD)	58.3 (4.3)	-	-	-	62.1 (4.8)	-	0.034
Inf.Vol. OL-HDF (L)-mean (SD)	27.1 (2.1)	-	-	-	28.1 (2.8)	-	0.102
QB (mL/min)-mean (SD)	342.4 (15.3)	-	-	-	347.0 (13.8)	-	0.667
APF (mL/min)-mean (SD)	−129.8 (−47.6)	-	-	-	−118.3 (39.2)	-	0.248
VPF (mL/min)-mean (SD)	171.4 (21.6)	-	-	-	173.1 (24.5)	-	0.999
SBP (mmHg)-mean (SD)	147.86 (13.95)	139.86 (16.54)	139.71 (9.52)	135 (8)	133 (12.41)	-	0.015
DBP (mmHg)-mean (SD)	78.29 (5.25)	72.86 (8)	72.29 (5.52)	69.71 (6.65)	69.71 (6.65)	-	0.041
Cr serum (mg/dL)-mean (SD)	4.80 (1.75)	4.43 (1.79)	4.75 (1.79)	4.90 (1.86)	5.01 (2.51)	0.517	0.735
U serum (mg/dL)-mean (SD)	142 (26.29)	118 (16.27)	130 (32.90)	127 (33.10)	128 (34.51)	0.360	0.128
Albumine (g/dL)-mean (SD)	3.23 (0.42)	3.27 (0.65)	3.42 (0.41)	3.18 (1.04)	3.51 (0.48)	0.211	0.063
Potassium serum (mmol/l)-mean (SD)	5.45 (0.43)	5.07 (0.43)	4.82 (0.30)	4.61 (0.33)	4.62 (0.29)	-	0.006
Ca serum (mg/dL)-mean (SD)	9.0 (0.30)	8.98 (0.29)	8.80 (0.10)	8.93 (0.19)	8.90 (0.19)	-	0.42
P serum (mg/dL)-mena (SD)	5.70 (0.39)	5.31 (0.29)	4.81 (0.37)	4.71 (0.56)	4.68 (0.39)	-	0.001
25OHD serum (ng/mL)-mean (SD)	18.94 (4.85)	20.77 (3.39)	24.24 (4.06)	24.71 (3.45)	27.08 (2.02)	-	0.007
PTH serum (pg/mL)-mean (SD)	197.98 (121)	228.1 (70.33)	280 (106.7)	253 (97.16)	241.1 (82.63)	-	0.122
U.A. serum (mg/dL)-mean (SD)	7.67 (1.95)	6.69 (0.54)	6.20 (0.70)	5.80 (0.52)	5.96 (0.52)	<0.001	0.018
HbA1c (g/dL)-mean (SD)	6.79 (0.68)	6.47 (0.63)	6.47 (0.59)	6.39 (0.54)	6.36 (0.59)	0.004	0.018
UACR (mg/g)-mean (SD)	4040 (2729)	2229 (1430)	1977 (1413)	1718 (1037)	1568 (746)	0.128	0.016
Glucosuria (mg/dL)-mean (SD)	13.71 (26.52)	229.14 (92.26)	277.57 (89.31)	292.85 (112.16)	286.14 (79.32)	-	0.001
KrU (mL/min)-mean (SD)	4.91 (1.14)	5.46 (1.71)	7.53 (1.76)	6.97 (1.50)	7.28 (1.68)	0.002	0.028
24 h CrCl (mL/min)-mean (SD)	12.7 (3.53)	16.43 (6.58)	17.94 (6.04)	16.29 (6.34)	16.35 (6.85)	0.180	0.236
Residual diuresis (mL/24 h)-mean (SD)	1742 (288)	2114 (533)	2135 (655)	1921 (410)	2021 (532)	0.305	0.546
Proteinuria (g/24 h)-mean (SD)	4.10 (1.95)	2.83 (1.68)	2.60 (1.86)	2.08 (1.17)	1.88 (0.71)	0.019	0.028

SD: standard deviation; N: number of patients; BMI: body mass index; ECW/TBW: extracellular water to total body water ratio; LUS: lung ultrasound; PV: portal vein; PF: pulsatility fraction; HVF: hepatic vein flow; S: S-wave, D: D-wave; IDWG: interdialytic weight gain; Inf. Vol. OL-HDF: infusion volume in post-dilutional online hemodiafiltration; QB: blood flow rate; APF: arterial pressure flow; VPF: venous pressure flow; SBP: systolic blood pressure; DBP: diastolic blood pressure; Cr: creatinine; U: urea; U.A.: uric acid; Ca: calcium; P: phosphorus; 25OHD: 25-hydroxyvitamin D; PTH: parathyroid hormone; HbA1c: hemoglobin A1c; UACR: urine albumin-to-creatinine ratio; KuR: residual kidney urea clearance; CrCl, creatinine clearance in 24 h. * Friedman test for multiple dependent variables. ** Wilcoxon test for two dependent variables (Baseline vs. 12-month follow-up).

## Data Availability

No new data were created or analyzed in this study. The data used to support the findings of this study are available from the corresponding author on request (Contact J.C.D.L.F., josedelaflor81@yahoo.com or jflomer@mde.es). I confirm that all the figures and tables are the original work of this manuscript’s authors. All have been performed by the authors of this manuscript, have not been adapted from other authors, and do not present an online link.

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
