# Peer review of "Efficacy and Safety of the Use of SGLT2 Inhibitors in Patients on Incremental Hemodialysis: Maximizing Residual Renal Function, Is There a Role for SGLT2 Inhibitors?"

_biomedicines, 2023, doi:10.3390/biomedicines11071908_

Round 1
Reviewer 1 Report
In the manuscript entitled “Efficacy and safety of the use of SGLT2 inhibitors in patients on incremental hemodialysis. Maximizing residual renal function, Is there a role for SGLT2 inhibitors?”, the authors evaluated the safety and efficacy of SGLT2i in seven patients on chronic hemodialysis and type 2 diabetes mellitus. Their main findings included an increase in KrU and urine output alongside a better control of extracellular volume, blood pressure, and glycemic control. The study is sound and contributes to knowledge in the field. Before we proceed, please address the following comments.
Comments
1) In the Methods section, the authors describe that 5 patients used Dapagliflozin and 2 patients Empagliflozin, although in Table 1, there are 6 patients being treated with Dapagliflozin and only 1 being treated with Empagliflozin. Correct the information accordingly.
2) Likewise, why only 7 out of 12 patients started treatment with iSGLT2 and not all of them? Did the other individuals refuse to sign the informed consent form or was the KrU too low?
3) What is the time of diabetes mellitus history before starting the hemodialysis?
4) Hyperkalemia is one of the “Achilles’ heel” in patients with end-stage kidney disease. Please provide information on potassium management following SGLT2i treatment as urine output is increased and is anticipated to also lead a decrease in potassium levels.
5) In addition, provide information on calcium, phosphorous, alkaline phosphatase, vitamin D and parathyroid hormone. There is emerging evidence on the effect of SGLT2i on modulating these parameters.
6) Provide information on loop diuretic use (dose and frequency) throughout the study. Was the dose maintained or decreased after treatment with SGLT2i?
Author Response
COVER LETTER RESPONDE TO REVIEWERS
Editor-in-Chief ¨Biomedicines¨: Prof. Dr. Shaker A. Mousa
Dear Assigned Editor Prof. Dr. Loredana Monica Demeny and Reviewers:
Thank you so much for your constructive suggestions and comments on our manuscript.
Please find enclosed our revised Manuscript ID biomedicines-2437680 entitled “Efficacy and safety of the use of SGLT2 inhibitors in patients on incremental hemodialysis. Maximizing residual renal function, Is there a role for SGLT2 inhibitors?" which we would like to resubmit for publication in Biomedicines.
The reviewer´s comments are all valuable and very helpful for improving our paper. We have studied these comments carefully and have made corrections accordingly, which we believe to have significantly improved the manuscript, and we hope that those changes will be met by your approval.
In the following pages, you will find our point-to-point responses to each of the reviewers’ Comments:
Reviewer 1 report:
Comments to authors
In the manuscript entitled “Efficacy and safety of the use of SGLT2 inhibitors in patients on incremental hemodialysis? Maximizing residual renal function, Is there a role for SGLT2 inhibitors?” the authors evaluated the safety and efficacy of SGLT2i in seven patients on chronic hemodialysis and type 2 diabetes mellitus. Their main findings included an increase in KrU and urine output alongside a better control of extracellular volume, blood pressure, and glycemic control. The study is sound and contributes to knowledge in the field. Before we proceed, please address the following comments.
Comments
1) In the Methods section, the authors describe that 5 patients used Dapagliflozin and 2 patients Empagliflozin, although in Table 1, there are 6 patients being treated with Dapagliflozin and only 1 being treated with Empagliflozin. Correct the information accordingly.
Thanks for the correction, indeed there are 5 patients with Dapagliflozin and 2 patients with Empagliflozin. The changes have been made table 1.
2) Likewise, why only 7 out of 12 patients started treatment with iSGLT2 and not all of them? Did the other individuals refuse to sign the informed consent form or was the KrU too low?
Thank you very much for your question. Our study is a case series with retrospective data collection of patients in our unit. Therefore, of the 12 diabetic patients who started incremental hemodialysis, only 7 of them started treatment with SGLT2i by indication of their treating nephrologist.
3) What is the time of diabetes mellitus history before starting the hemodialysis?
Thanks for the relevant and important question. We have added a row on table 1 with the time in years of T2DM history.
4) Hyperkalemia is one of the “Achilles’ heel” in patients with end-stage kidney disease. Please provide information on potassium management following SGLT2i treatment as urine output is increased and is anticipated to also lead a decrease in potassium levels.
Thank you for the recommendation. We will add to Table 2 and on the results (line 182) the mean follow-up potassium values of the 7 patients during the follow-up period. We will also add a commentary of this in the discussion (Lines 336-344, Luo X, Xu MM, Jing, Zhou S, Xue MM, Cheng, et al. Influence of SGLT2i and RAASi and Their Combination on Risk of Hyperkalemia in DKD: A Network Meta-Analysis. Clinical Journal of the American Society of Nephrology. :10.2215/CJN.0000000000000205).
5) In addition, provide information on calcium, phosphorous, alkaline phosphatase, vitamin D and parathyroid hormone. There is emerging evidence on the effect of SGLT2i on modulating these parameters.
Great recommendation. We have added on Table 2, the mean values of calcium, phosphorus, vitamin D and PTH during follow-up. We do not add alkaline phosphatase because we do not usually measure it in hemodialysis patients. We also added a brief paragraph in the discussion (lines 346-351).
6) Provide information on loop diuretic use (dose and frequency) throughout the study. Was the dose maintained or decreased after treatment with SGLT2i?
Thanks for the recommendation, you can find on table 1 the furosemide dose per day. The addition of SGLT2i increased mean diuresis by 279 ml/day, as we performed periodic controls of bio-impedance and intravascular congestive parameters by lung ultrasound and VEXUS score (without pulsed Doppler of the renal veins), thus adjusting the dry weights and avoiding hypotensive episodes. Since there were no significant changes in volemia, we did not modify the furosemide regimen in the 7 patients during the follow-up period. We added a comment in the methodology about the absence of changes in the furosemide dose during follow-up.
Reviewer 2 Report
This interesting and well written retrospective study relates the beneficial effects of Glifozins in 7 diabetic patients treated by incremental hemodialysis in Spain. The data are convincing and the discussion includes an analysis of all publications on the topic.
The quality of English wording seems excellent
Author Response
COVER LETTER RESPONDE TO REVIEWERS
Editor-in-Chief ¨Biomedicines¨: Prof. Dr. Shaker A. Mousa
Dear Assigned Editor Prof. Dr. Loredana Monica Demeny and Reviewers:
Thank you so much for your constructive suggestions and comments on our manuscript.
Please find enclosed our revised Manuscript ID biomedicines-2437680 entitled “Efficacy and safety of the use of SGLT2 inhibitors in patients on incremental hemodialysis. Maximizing residual renal function, Is there a role for SGLT2 inhibitors?" which we would like to resubmit for publication in Biomedicines.
The reviewer´s comments are all valuable and very helpful for improving our paper. We have studied these comments carefully and have made corrections accordingly, which we believe to have significantly improved the manuscript, and we hope that those changes will be met by your approval.
In the following pages, you will find our point-to-point responses to each of the reviewers’ Comments:
Reviewer 2 report:
Comments to authors
This interesting and well-written retrospective study relates the beneficial effects of Glifozins in 7 diabetic patients treated by incremental hemodialysis in Spain. The data are convincing and the discussion includes an analysis of all publications on the topic.
Thank you for the kind comment; it encourages us to continue studying the renal benefits of SGLT2i in the hemodialysis population with residual renal function.